# Calculation and Analysis of Pipe Joint Settlement Control in Large Back Silting Immersed Tube Tunnel

**Zhijun Li [1], Xiabing Yue [2],\* and Guanqing Wu [2],\***

1    Shanxi Transportation Construction Project Supervision & Consulting Group Co., Ltd., Taiyuan 030006, China
2    Highway of School, Chang'an University, Xi'an 710064, China
\*    Correspondence: yuexb@chd.edu.cn (X.Y.); wogting@163.com (G.W.)

**Abstract:** The use of the segmental pipe section immersed tunnel suffers from several problems, such as complex construction, weak foundation, great water depth, great thickness of siltation back on the top of the tube, and difficult settlement control. Based on Winkel's elastic foundation beam theory, a mechanical calculation model is established according to the case of an inhomogeneous soil layer, and the force and deformation of the structural system of the immersed tube tunnel are calculated based on a bridge in Zhuhai as an example of an immersed tube tunnel. The results show that the derived formula for calculating the allowable differential settlement per unit length of the longitudinally immersed tube is applicable to the sudden change type foundation stiffness deformation model of the natural foundation section of the tube tunnel of the aforementioned bridge in Zhuhai. The relationship between the settlement control index and related influencing factors is analyzed. Hence, a formula for calculating the stratified ground foundation's integrated bed coefficient is derived, and the equations for tunnel deflection curves and shear forces are solved. A set of calculation methods that are applicable to the foundation settlement control criteria of the segmental immersed tunnel is provided, and the results provide a significant reference for the optimization of the foundation scheme and improvement of the construction process for similar projects.

**Keywords:** immersed tube tunnel; large back-silt soft foundation; segmental pipe section; differential settlement; calculation analysis

## 1. Introduction

Since the middle and late 1970s, China began to study immersed tunnels and has made substantial achievements [1–6]. Since then, immersed tunnels have made significant progress in China [7–10]. Note that immersed tunnels are widely used in cross-sea engineering with the advantages of a short construction period, high waterproof ability, and strong adaptability [11–14]. Previous studies on tunnel settlement control mainly focus on the shield tunnel. On the other hand, there is little research on the settlement control standard of immersed tunnels as they are complicated. Moreover, the majority of immersed tunnels in China are deep-buried tunnels and segmental pipe joints are highly affected by the cyclic load of back silting and desilting during the operation period.

From the essence of pipe joint displacement in immersed tube tunnels, the settlement is mainly caused by the compression deformation of the underlying base layer of the tunnel; thus, it is necessary to discuss the settlement control standard. Hu et al. [15] studied the settlement control standard of the segmental joint shear key of immersed tunnels using large-scale model tests and finite elements and concluded that under the longitudinal and transverse differential settlement, the stress process of the segmental joint shear key is able to be divided into three stages: a compression stage, a stress stage, and a yield state. Xu et al. [16] investigated the settlement control of the immersed tube tunnel foundation at the junction of an artificial island and tunnel via laboratory tests and field tests and obtained the reinforcement measures of the immersed tube tunnel foundation

at the junction of the island and the tunnel and the pre-lift value of the pipe section. Niu et al. [17,18] studied the deformation characteristics and settlement characteristics of the segmental pipe section joint of the immersed tube structure using a model test and a finite element model and acquired the influence of the settlement and opening of the immersed tube tunnel, the stress–strain characteristics of the segment joint, and the compressive stress distribution of marine subsoil. The increase in prestress effectively restrains the settlement of the pipe section joint element and the joint's expansion. Xue et al. [19] took the Hong Kong–Zhuhai–Macao immersed tunnel as an example and presented a calculation of the reasonable buried depth for an immersed tunnel. Yue et al. [20] considered the section of the natural foundation of the immersed tube tunnel of the Hong Kong–Zhuhai–Macao Bridge as the research object, carried out centrifugal model tests, and obtained the resilience and recompression characteristics of the natural foundation of the immersed tube tunnel, as well as the deformation characteristics of the immersed tube structure. Wang et al. [21,22] introduced several ground treatment methods used in the Hong Kong–Zhuhai–Macao Bridge project to reduce the potential uneven ground settlement between artificial islands and submerged tunnels. One of the foundation improvement methods involved a combination of preloaded and high-pressure jet grouting (HPJG) columns. Furthermore, they introduced the settlement characteristics of the immersed tube tunnel in the traffic engineering of the Hong Kong–Zhuhai–Macao Bridge. Li et al. [23] proposed a comprehensive prediction model combining time-series decomposition, the least square method, the sparrow search algorithm (SSA), and the support vector regression (SVR) and predicted the thorough deformation of the immersed tube tunnel in operation using the deformation monitoring data of the E13–E14 and E17–E18 of the Hong Kong–Zhuhai–Macao Bridge.

In conclusion, based on the elastic foundation beam theory, this study starts with the structural system of immersed tube tunnels and subsequently establishes a mechanical model considering uneven soil layers. The Winkler model presents a foundation as an arrangement of a limited number of soil columns that have been divided [24]. It was depicted by a sequence of independent springs that are responsible for supporting the load P at a particular point on a proximate base surface of a superstructure. These springs worked together to balance the load P with their respective forces. However, since the springs operated independently of each other, they only provided local resistance at the point of loading and failed to generate resistance elsewhere [25]. Numerous achievements were proposed based on this model [26–33], deducing and calculating the stress and deformation of the structural system of immersed tube tunnels under the action of uneven soil layers and analyzing the sensitivity of various influencing factors to the settlement, in addition to obtaining a set of reasonable calculation methods for settlement control standards of segmental immersed tube tunnels. The research results provide a favorable scientific basis for the optimization of the tunnel foundation scheme and the improvement of the construction technology to ensure the safety of the structure to a greater extent.

## 2. Foundation Beam Model of Uneven Soil Layer

The Zhuhai immersed tube tunnel project, which passes through five different strata, is the first of its kind in the tunnel industry, domestically and internationally. The structure system of the immersed tube tunnel is composed of the immersed pipe segment, segment joint, pipe joint, and shock absorption cable. There is a nonlinear effect between the immersed tube tunnel structure and foundation stiffness; thus, it is difficult to use static analysis. As for the immersed tube tunnel, the mechanical properties of its structural system are the main factors in the settlement control standard research. Therefore, the settlement control standard research of immersed tube tunnels mainly revolves around the high-order statically indeterminate problem of the interaction between the immersed tube tunnel structural system and the foundation soil and backfill soil.

### 2.1. Winkler Foundation Model

According to the Winkler foundation model, the pressure strength at any point on the foundation is:

$$p = ks \tag{1}$$

where $k$ is the coefficient of the foundation bed; $p$ denotes the pressure intensity (kPa); $s$ represents the settlement deformation of foundation (m).

The mechanical model of a constant cross-section infinite beam with a width $b$, placed on a Winkler elastic foundation, is shown in Figure 1. A micro section with a length dx along the longitudinal direction of the beam is taken for force analysis. Under the action of a uniform load $q$, the shear force $Q$ and bending moment $M$ of the micro section are as follows:

$$\frac{dM}{dx} = Q, \ \frac{dQ}{dx} = pb - qb \tag{2}$$

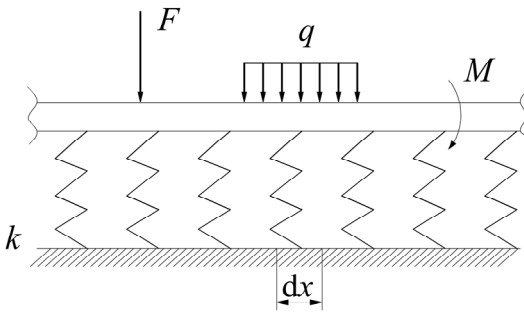

**Figure 1.** Elastic foundation beam model and stress state.

Subsequently, the flexural differential equation of the beam section is obtained as follows:

$$EI\frac{d^2w}{dx^2} = -M \tag{3}$$

where $E$ is the elastic modulus and $I$ the moment of inertia of the section.

According to Equations (2) and (3):

$$EI\frac{d^4w}{dx^4} = qb - pb \tag{4}$$

For beams on an elastic foundation, the deflection $w$ is equal to $s$. Hence, $p = ks = kw$. Substitute in Equation (4) to obtain:

$$EI\frac{d^4w}{dx^4} = qb - kwb \tag{5}$$

According to Equation (3), the bending moment on this section is $M = -EI\frac{d^2w}{dx^2}$.

Combined with Equation (2), the shear force on this section is $Q = \frac{dM}{dx} = -EI\frac{d^3w}{dx^3}$, and the angle is $\theta = \frac{dw}{dx}$.

Take the concentrated force $F$ at the midpoint of the elastic foundation beam, as shown in Figure 2, and calculate the left half; then, according to Equation (5), the deflection $w$ is:

$$w = e^{\lambda x}(c_1 \cos \lambda x + c_2 \sin \lambda x) + e^{-\lambda x}(c_3 \cos \lambda x + c_4 \sin \lambda x) \tag{6}$$

where $\lambda = \sqrt[4]{\frac{kb}{4EI}}$; $c_1$, $c_2$, $c_3$, and $c_4$ are undetermined constants.

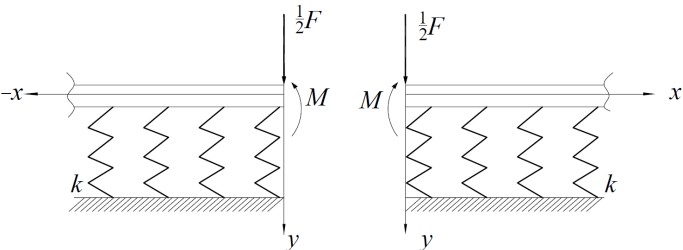

**Figure 2.** Concentrated load foundation beam model and stress state.

According to the calculation in Figures 1 and 2:

When $x$ approaches infinity, the deflection $w$ tends to 0, and by substituting into Equation (6), $c_1 = c_2 = 0$.

When $x$ is at the origin O, the angle is $\theta = \frac{dw}{dx} = 0$, and the shear force is $Q = -EI\frac{d^3w}{dx^3} = -\frac{F}{2}$, thus, $c_1 = c_2 = \frac{F}{8\lambda^3 EI}$.

Substituting the assumed coefficient into Equation (6) yields:

$$w = \frac{F\lambda}{2kb}e^{\lambda x}(\cos\lambda x + \sin\lambda x)$$
$$= \frac{F}{8\lambda^3 EI}e^{-\lambda x}(\cos\lambda x + \sin\lambda x) \tag{7}$$

*2.2. Foundation Beam Model under Concentrated Load*

2.2.1. The Concentrated Load Acts on the Boundary of the Soil Layer

As shown in Figure 3, the concentrated load acts on point O, and $k_1$ and $k_2$ are the foundation bed coefficients acting on the left and right of the origin O, respectively.

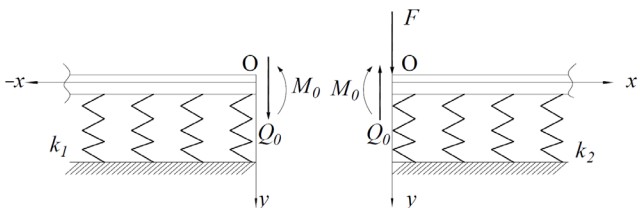

**Figure 3.** Foundation beam model and stress under concentrated load.

The homogeneous solution of the control Equation (5) of the Z-section is Equation (6), and the general solution of the left section beam of $k_1$ is obtained as follows:

$$
\begin{aligned}
w_1 = \; & e^{\lambda_1 x}(c_{11}\cos\lambda_1 x + c_{21}\sin\lambda_1 x) \\
& + e^{-\lambda_1 x}(c_{31}\cos\lambda_1 x + c_{41}\sin\lambda_1 x)
\end{aligned} \tag{8}
$$

where $\lambda_1 = \sqrt[4]{\frac{k_1 b}{4EI}}$. When $x$ tends to the left end of infinity, the deflection tends to 0; while at the origin O, the bending moment is $M = M_0$, and the shear force is $Q = Q_0$, which is obtained by introduction into the general solution of Equation (8):

$$c_{11} = \frac{Q_0 - \lambda_1 M_0}{2\lambda_1^3 EI}, \; c_{21} = \frac{-M_0}{2\lambda_1^2 EI}, \; c_{31} = c_{41} = 0$$

By introducing Equation (8), the following equation is acquired:

$$w_1 = \frac{e^{\lambda_1 x}}{2\lambda_1^3 EI}[Q_0\cos\lambda_1 x - \lambda_1 M_0(\cos\lambda_1 x + \sin\lambda_1 x)] \tag{9}$$

Therefore, the following is obtained: $w_1 = \frac{Q_0 - \lambda_1 M_0}{2\lambda_1^3 EI}$, $\theta_1 = \frac{Q_0 - 2\lambda_1 M_0}{2\lambda_1^2 EI}$.

Similarly, the general solution of the beam at the right end of the foundation bed coefficient $k_2$ is:

$$w_2 = \frac{e^{-\lambda_2 x}}{2\lambda_2{}^3 EI}[(F - Q_0)\cos \lambda_2 x - \lambda_2 M_0 \cos \lambda_2 x + \lambda_2 M_0 \sin \lambda_2 x] \tag{10}$$

$$w_2 = \frac{F - Q_0 - \lambda_2 M_0}{2\lambda_2{}^3 EI}, \quad \theta_2 = \frac{Q_0 - F + 2\lambda_2 M_0}{2\lambda_2{}^2 EI}.$$

Based on the fact that the deflection of the beam segment at the origin O is equal to the rotation angle, the deflection of the beam segment at the origin is equal to the angle of rotation.

$$\frac{F - Q_0 - \lambda_2 M_0}{2\lambda_2{}^3 EI} = \frac{Q_0 - \lambda_1 M_0}{2\lambda_1{}^3 EI}, \quad \frac{Q_0 - F + 2\lambda_2 M_0}{2\lambda_2{}^2 EI} = \frac{Q_0 - 2\lambda_1 M_0}{2\lambda_1{}^2 EI}$$

Hence, it can be obtained that:

$$M_0 = \frac{\lambda_1 \lambda_2 F}{\lambda_1{}^3 + \lambda_1 \lambda_2{}^2 + \lambda_1{}^2 \lambda_2 + \lambda_2{}^3}, \quad Q_0 = \frac{\lambda_1{}^2 F}{\lambda_1{}^2 + \lambda_2{}^2} \tag{11}$$

### 2.2.2. The Applied Point of Concentrated Load Deviates from the Soil Boundary Point

The foundation soil layer under the beam is inhomogeneous, O is the coordinate origin, $k_1$ and $k_2$ are, respectively, the left and right foundation bed coefficients, and the concentrated load $F$ acts on the right beam, with the distance $L$, as shown in Figure 4.

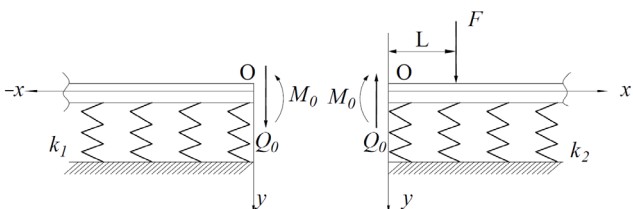

**Figure 4.** Foundation beam model and stress state under eccentric concentrated load.

For the beam model with a foundation bed coefficient $k_1$ in the left section, the deflection curve equation of this section is acquired based on Equation (9), as follows:

$$w_1 = \frac{e^{\lambda_1 x}}{2\lambda_1{}^3 EI}[Q_0 \cos \lambda_1 x - \lambda_1 M_0(\cos \lambda_1 x + \sin \lambda_1 x)] \tag{12}$$

As for the beam model with the foundation bed coefficient $k_2$ in the right section, the deflection equation at the section $x \neq L$ is obtained from Equation (7):

$$w_{21} = \frac{F}{8\lambda_2{}^3 EI} e^{-\lambda_2 |x - L|}(\cos \lambda_2 |x - L| + \sin \lambda_2 |x - L|) \tag{13}$$

According to $M = -EI\frac{\mathrm{d}^2 w}{dx^2}$ and $Q = \frac{dM}{dx} = -EI\frac{\mathrm{d}^3 w}{dx^3}$, the shear force and bending moment on the section are calculated as:

$$M_L = \frac{F}{4\lambda_2} e^{-\lambda_2 L}(\cos \lambda_2 L - \sin \lambda_2 L) \tag{14}$$

$$Q_L = \frac{F}{2} e^{-\lambda_2 L} \cos \lambda_2 L \tag{15}$$

However, the bending moment and shear value of the right beam section at the section at point O are not the results of Equations (14) and (15), where the actual force couple is

$M_0 - M_L$, and the applied force is $Q_L - Q_0$. The flexural curve equation of the left beam is obtained from Equation (10):

$$w_{22} = \frac{e^{-\lambda_2 x}}{2\lambda_2{}^3 EI}[(Q_L - Q_0)\cos \lambda_2 x - \lambda_2(M_0 - M_L) \cdot (\cos \lambda_2 x - \sin \lambda_2 x)] \tag{16}$$

The superposition of Equations (13) and (16) is the deflection curve equation of the right beam under the actual condition:

$$\begin{aligned} w_2 = \ & w_{21} + w_{22} = \frac{F}{8\lambda_2{}^3 EI} e^{-\lambda_2|x-L|}(\cos \lambda_2|x-L| + \sin \lambda_2|x-L|) \\ & + \frac{e^{-\lambda_2 x}}{2\lambda_2{}^3 EI}[(Q_L - Q_0)\cos \lambda_2 x - \lambda_2(M_0 - M_L) \cdot (\cos \lambda_2 x - \sin \lambda_2 x)] \end{aligned} \tag{17}$$

Taking the derivative of this equation yields the corresponding deflection and angle of rotation. Hence, the deflection at the origin of the left and right beam segments is, respectively:

$$w_1 = \frac{Q_0 - \lambda_1 M_0}{2\lambda_1{}^3 EI}, \ w_2 = \frac{e^{-\lambda_2 L}F(\sin \lambda_2 L + \cos \lambda_2 L) + 4(Q_L - Q_0) - 4\lambda_2(M_0 - M_L)}{8\lambda_2{}^3 EI}$$

The angle of rotation at the origin of the left and right beam segments are, respectively:

$$\theta_1 = \frac{Q_0 - 2\lambda_1 M_0}{2\lambda_1{}^2 EI}, \ \theta_2 = \frac{e^{-\lambda_2 L}F\sin \lambda_2 L + 2(Q_0 - Q_L) + 4\lambda_2(M_0 - M_L)}{4\lambda_2{}^2 EI}$$

The shear force $Q_0$ and bending moment $M_0$ at the section are obtained from the left and right deflection and angle:

$$\begin{aligned} Q_0 &= \frac{F\lambda_1{}^2 e^{-\lambda_2 L}[\lambda_1(\sin \lambda_2 L + \cos \lambda_2 L) + \lambda_2(\cos \lambda_2 L - \sin \lambda_2 L)]}{\lambda_1{}^3 + \lambda_1{}^2 \lambda_2 + \lambda_2{}^3 + \lambda_1 \lambda_2{}^2} \\ M_0 &= \frac{F\lambda_1 e^{-\lambda_2 L}[\lambda_2(\lambda_1 \sin \lambda_2 L + \lambda_2 \cos \lambda_2 L) - (\lambda_1{}^2 + \lambda_2{}^2)\sin \lambda_2 L]}{\lambda_1{}^3 \lambda_2 + \lambda_1{}^2 \lambda_2{}^2 + \lambda_2{}^4 + \lambda_1 \lambda_2{}^3} \end{aligned} \tag{18}$$

Similarly, a concentrated force $F_1$ is applied at $L_1$ from the left beam segment to the foundation junction section, and a concentrated force $F_2$ is applied at $L_2$ from the right beam segment to the foundation junction section, as shown in Figure 5.

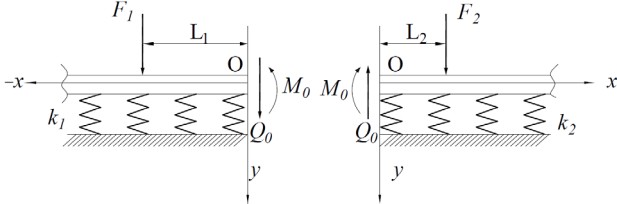

**Figure 5.** Foundation beam model and stress state under two concentrated loads.

According to the results obtained in Equation (18), the deflection equations of the left and right beams are acquired as follows:

$$\begin{aligned} w_1 = \ & w_{11} + w_{12} = \frac{F}{8\lambda_1{}^3 EI} e^{-\lambda_1|x+L_1|}(\cos \lambda_1|x+L_1| + \sin \lambda_1|x+L_1|) \\ & + \frac{e^{\lambda_1 x}}{2\lambda_1{}^3 EI}[(Q_0 - Q_L)\cos \lambda_1 x - \lambda_1(M_0 - M_L) \cdot (\cos \lambda_2 x + \sin \lambda_2 x)] \\ w_2 = \ & w_{21} + w_{22} = \frac{F}{8\lambda_2{}^3 EI} e^{-\lambda_2|x-L_2|}(\cos \lambda_2|x-L_2| + \sin \lambda_2|x-L_2|) \\ & + \frac{e^{-\lambda_2 x}}{2\lambda_2{}^3 EI}[(Q_L - Q_0)\cos \lambda_2 x - \lambda_2(M_0 - M_L) \cdot (\cos \lambda_2 x - \sin \lambda_2 x)] \end{aligned} \tag{19}$$

The deflection and angle of the left and right sections at the foundation boundary are obtained by differentiating Equation (16). The equation of shear force and bending moment at the boundary section of the inhomogeneous foundation is obtained based on the fact that the deflection and angle of left and right are, respectively, equal.

### 2.3. Foundation Beam Model under Uniform Load

Based on Equation (19), the deflection curve equation is obtained:

$$
\begin{aligned}
w_1 = \ & w_{11} + w_{12} = \int_0^L \frac{q}{8\lambda_2{}^3 EI} e^{-\lambda_2 |x+y|}(\cos \lambda_2|x+y| + \sin \lambda_2|x+y|)\mathrm{dy} \\
& + \frac{e^{\lambda_1 x}}{2\lambda_1{}^3 EI}[(Q_0 + Q_1)\cos \lambda_1 x - \lambda_1(M_0 - M_1) \cdot (\cos \lambda_1 x + \sin \lambda_1 x)] \\
w_2 = \ & w_{21} + w_{22} = \int_0^L \frac{q}{8\lambda_2{}^3 EI} e^{-\lambda_2 |x-y|}(\cos \lambda_2|x-y| + \sin \lambda_2|x-y|)\mathrm{dy} \\
& + \frac{e^{-\lambda_2 x}}{2\lambda_2{}^3 EI}[(Q_2 - Q_0)\cos \lambda_2 x - \lambda_2(M_0 - M_2) \cdot (\cos \lambda_2 x - \sin \lambda_2 x)]
\end{aligned}
\tag{20}
$$

The shear force $Q_0$ and bending moment $M_0$ at the boundary section of the inhomogeneous foundation are acquired from the continuous equality of beam deflection and rotation angle at the boundary section:

$$
Q_0 = \frac{A}{-\lambda_1 \lambda_2 (2\lambda_1{}^2 \lambda_2{}^2 + 2\lambda_1 \lambda_2{}^3 + 2\lambda_1{}^3 \lambda_2 + \lambda_1{}^4 + \lambda_2{}^4)}
\tag{21}
$$

$$
M_0 = \frac{B}{-\lambda_1 \lambda_2 (2\lambda_1{}^2 \lambda_2{}^2 + 2\lambda_1 \lambda_2{}^3 + 2\lambda_1{}^3 \lambda_2 + \lambda_1{}^4 + \lambda_2{}^4)}
\tag{22}
$$

where:

$$
\begin{aligned}
A = \ & q\lambda_1{}^4(\lambda_1 + \lambda_2)\left[e^{-\lambda_2 L}(\cos \lambda_2 L - \sin \lambda_2 L) - 1\right] - \tfrac{1}{2}q\lambda_2{}^3(\lambda_1{}^2 - \lambda_2{}^2)\left[1 - e^{-\lambda_1 L}(\cos \lambda_1 L + \sin \lambda_1 L)\right] \\
& + q\lambda_1{}^3 e^{-\lambda_2 L}(\lambda_1{}^2 - \lambda_2{}^2)\sin \lambda_2 L
\end{aligned}
$$

$$
B = \frac{q\lambda_2{}^2(\lambda_1{}^3 + \lambda_2{}^3)\left[1 - e^{-\lambda_1 L}(\cos \lambda_1 L + \sin \lambda_1 L)\right]}{2\lambda_1} + \frac{q\lambda_1{}^3(\lambda_1{}^2 - \lambda_2{}^2)\left[1 - e^{-\lambda_2 L}(\cos \lambda_2 L - \sin \lambda_2 L)\right]}{2\lambda_2} - \frac{q\lambda_1{}^2 e^{-\lambda_2 L}(\lambda_1{}^3 + \lambda_2{}^3)e^{-\lambda_1 L}\sin \lambda_2 L}{\lambda_2}
$$

### 2.4. The Method of Determining the Foundation Bed Coefficient k

The foundation bed coefficient $k$ of the immersed tube tunnel foundation should be taken as the average foundation bed coefficient of each soil layer, which is equal to the ratio of the additional stress $p$ of the foundation and the average settlement $s$ of the bottom of the immersed tube:

$$
k = \frac{p}{s}
\tag{23}
$$

It is defined that $S_1$ is the settlement amount of the gravel cushion after the immersed tube is backfilled, $S_2$ is the settlement deformation amount corresponding to the clay layer, and $S_3$ is the settlement deformation amount corresponding to the sand layer. Afterwards, the average foundation bed coefficient is expressed by the combination of the coefficient of each soil layer. Due to the special structure of the immersed tube tunnel, the additional stress of the base changes very little along the depth direction, so Equation (2) is expressed as:

$$
k = \frac{p}{s} = \frac{p}{S_1 + S_2 + S_3} = \frac{1}{\frac{S_1 + S_2 + S_3}{p}} = \frac{1}{\frac{1}{k_1} + \frac{1}{k_2} + \frac{1}{k_3}}
\tag{24}
$$

In this study, according to the field load plate test of the Hong Kong–Zhuhai–Macao immersed tube tunnel, the values of the foundation bed coefficients of each soil type are shown in Table 1.

**Table 1.** Values of subgrade bed coefficient of each soil type.

| Classification and Characteristics of Soil Layers | $k$ ($10^4$ kN/m³) | Classification and Characteristics of Soil Layers | $k$ ($10^4$ kN/m³) |
| --- | --- | --- | --- |
| Mucky soil | 0.1~0.5 | Dense sand or loose gravel | 2.5~4.0 |
| Soft clay | 0.5~1.0 | Dense gravel | 4.0~10 |
| Clay and silty clay (soft plastic) | 1.0~2.0 | Soft or moderately strong weathered hard rock | 20~100 |
| Clay and silty clay (plastic) | 2.0~4.0 | Hard rock | 100~150 |
| Clay and silty clay (hard plastic) | 4.0~10 | Block stone | 500~600 |
| Loose sand | 1.0~1.5 | Concrete, reinforced concrete | 800~1500 |
| Medium dense sand or loose gravel | 1.5~2.5 | / | / |

Due to the influence of the foundation's size and buried depth, the foundation bed coefficient $k$ needs to be modified to some extent. Assuming that the foundation width is $D$ and the foundation length is $L$, the corrected foundation bed coefficient $k'$ is:

Sandy soil: $k' = k \cdot \frac{D+0.305}{2D}$; Clayey soil; $k' = k \cdot \frac{0.305}{D}$; Affected by length: $k' = k \cdot \frac{2L+D}{3L}$.

## 3. Method of Settlement Control Calculation

The deflection and shear force of the beam model are solved using the elastic foundation beam theory under the concentrated force of shear force (18), and the influence line of shear along the longitudinal stiffness of the beam is drawn, as shown in Figure 6.

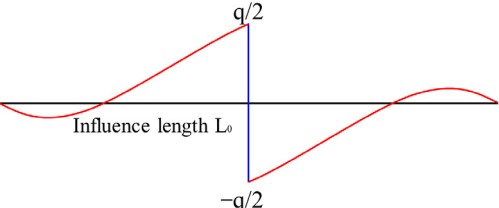

**Figure 6.** Stiffness influence line.

The distances between the two intersection points and the origin are equal in Figure 6, and the calculation formula is:

$$L = \frac{\pi}{2} \sqrt[4]{\frac{4EI}{\overline{k}}} \tag{25}$$

where $\overline{k}$ is the average value of the foundation bed stiffness in the range of $[-L, L]$, $\overline{k} = \frac{1}{r} \sum_{i=1}^{r} k_i$; $EI$ denotes the flexural stiffness.

According to the foundation stiffness distribution diagram (Figure 7), there are multiple variation modes in the foundation bed stiffness distribution of the natural foundation, under the condition of no construction deviation. The abrupt foundation bed stiffness distribution mode is the most unfavorable working condition, while the abrupt foundation bed stiffness is the most unfavorable stress position of the immersed tube structure.

Therefore, the average values $k_1$ and $k_2$ of the foundation bed stiffness of the left and right beam sections are, respectively, taken in its scope:

$$k_1 = \frac{1}{m} \sum_{i=1}^{m} k_i, \ k_2 = \frac{1}{n} \sum_{i=1}^{n} k_i \tag{26}$$

Thus, the distribution diagram of foundation stiffness variation is drawn, as shown in Figure 8.

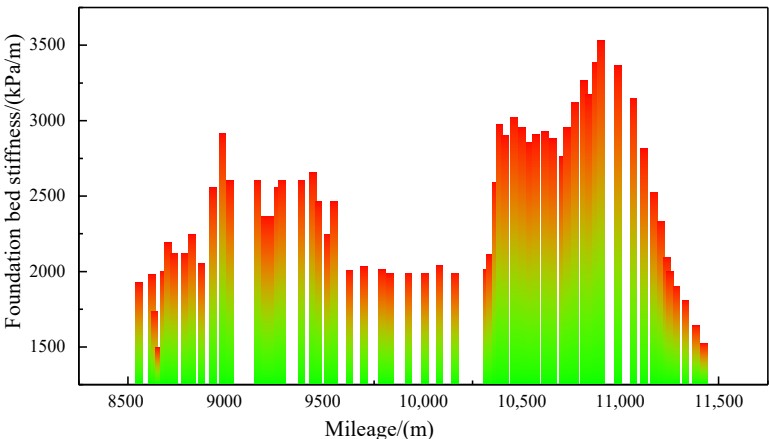

**Figure 7.** Distribution diagram of overall foundation stiffness of natural foundation.

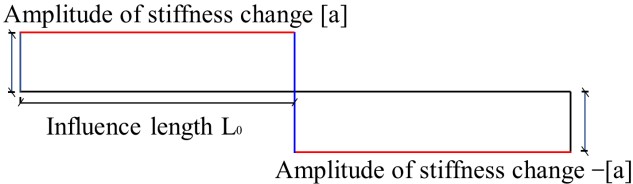

**Figure 8.** Distribution of foundation stiffness changes.

The overlying distributed load remains unchanged, and the beam model on uniform foundation stiffness produces a uniform settlement *s* under the action of the uniform distributed load, based on the previous assumptions. The graph multiplication calculation is carried out between Figures 6 and 8, and the relationship of the change rate between the allowable joint shear force and allowable foundation bed stiffness is obtained as follows:

$$[a] = \frac{6[Q]}{(1 + \lambda + \lambda^2) \cdot \bar{k} \cdot s \cdot L \cdot \kappa} \tag{27}$$

where $[Q]$ is the allowable shear value of the shear key; $[a]$ represents the amplitude of the foundation stiffness change rate; $\lambda$ denotes the change factor of the distribution pattern; $\kappa$ is the centroid equivalent factor; $s$ denotes the settlement value with average foundation stiffness.

The left and right stiffnesses of the beam segment within $2L$ are:

$$k_1 = (1 + [a]) \cdot \bar{k}, \ k_2 = (1 - [a]) \cdot \bar{k} \tag{28}$$

The left and right settlements of the beam section within $2L$ are:

$$s_1 = \frac{s}{1 + [a]}, \ s_2 = \frac{s}{1 - [a]} \tag{29}$$

Therefore, the allowable average differential settlement per unit length of the longitudinal beam is:

$$[\Delta] = \frac{S_2 - S_1}{2L} = \frac{S}{2L} \cdot \left( \frac{1}{1 - [a]} - \frac{1}{1 + [a]} \right) \tag{30}$$

Since the change in foundation stiffness is abrupt, $\lambda = 1$, substituting Equation (27) and $\lambda = 1$ into Equation (30), the following equation is obtained:

$$[\Delta] = \frac{2[Q]\bar{k}s^2\kappa}{\left(\bar{k}sL\kappa\right)^2 - 4[Q]^2} \tag{31}$$

According to the result of graph multiplication, the centroid equivalent factor is acquired: $\kappa = 2 \times \frac{1+e^{-\frac{\pi}{2}}}{\pi} = 0.77$.

The settlement $s_i$ at the junction of each segment composed of $n$ segments and the differential settlement along the longitudinal unit length between any two adjacent segments are:

$$\Delta_i = \frac{|s_{i+1} - s_i|}{L_i} \tag{32}$$

The maximum differential settlement in the whole section is obtained by calculating the average differential settlement per unit length between adjacent $n$ sections:

$$\max\Delta_i = \max\frac{|s_i - s_j|}{L_{ij}} \tag{33}$$

When $\max\Delta_i \leq [\Delta]$, the calculated longitudinal differential settlement meets the requirements of joint shear force.

When the overlying load in the study section changes slightly along the longitudinal direction, the calculation model is the non-uniform foundation stiffness model under a uniform load. The shear force calculated by Equation (21) meets the requirements when $Q \leq [Q]$ is satisfied.

## 4. Case

The cross-section width of the Zhuhai immersed tube tunnel is 37.95 m, the height is 11.4 m, and the bending stiffness is $EI = 9.7 \times 10^{10} \text{kN} \cdot \text{m}^2$. The length of each pipe section is 180 m, which is composed of eight C55 concrete sections connected by joints, and the length of each section is 22.5 m. According to the natural foundation stiffness distribution diagram of the Zhuhai Bridge (Figure 7), the section with abrupt foundation stiffness is selected as a calculation example, and the settlement control calculation method deduced in this study is adopted to analyze the settlement control calculation.

(1) According to the existing calculation results of the Zhuhai immersed tube tunnel, combined with the characteristics of the immersed tube shear bond and structural size, the allowable shear value at the joint is calculated to be $[Q] = 1.6 \times 10^4 \text{kN}$.

(2) The longitudinal distribution of the average load and settlement of the pipe bottom is shown in Figures 9 and 10. The foundation stiffness corresponding to each segment is calculated according to Equation (23), as shown in Table 2.

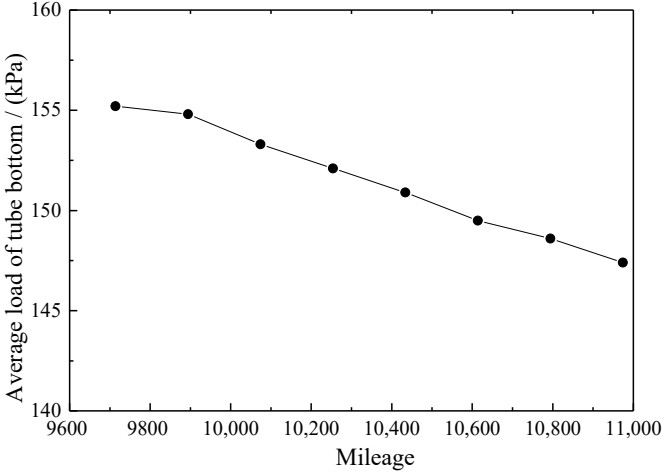

**Figure 9.** Pipe bottom load diagram in calculation scope.

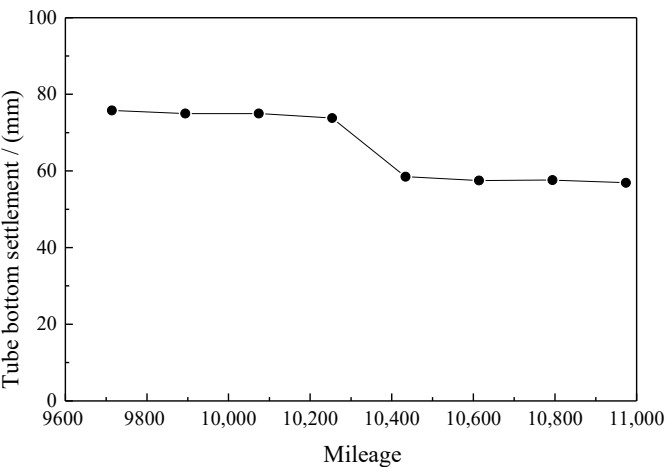

**Figure 10.** Foundation settlement within calculation scope.

**Table 2.** Stiffness solution table.

| Section No | 1 | 2 | 3 | 4 | 5 | 6 | 7 | 8 | Average |
|---|---|---|---|---|---|---|---|---|---|
| Average load kPa | 155.2 | 154.8 | 153.3 | 152.1 | 150.9 | 149.5 | 148.6 | 147.4 | 151.475 |
| Average settlement m | 0.0758 | 0.0750 | 0.0750 | 0.0738 | 0.0585 | 0.0575 | 0.0576 | 0.0569 | 0.0663 |
| Foundation stiffness $10^4$ kN/m$^2$ | 7.772 | 7.831 | 7.752 | 7.824 | 9.796 | 9.871 | 9.793 | 9.826 | 8.808 |

(3) The specific longitudinal length interval affected by the joint shear force is acquired from Equation (25):

$$L = \frac{\pi}{2} \sqrt[4]{\frac{4EI}{\bar{k}}} = 71.98 \text{ m}$$

(4) The average allowable differential settlement per unit length of the immersed tube in the range of length 2$L$ is calculated by Equation (31):

$$[\Delta] = \frac{2[Q]\bar{k}s^2\kappa}{\left(\bar{k}sL\kappa\right)^2 - 4[Q]^2} = 0.9197 \text{ mm/m}$$

(5) After checking the calculation, the actual settlement $s_i$ at any two points on the left and right ends of the joint section and the maximum value max$\Delta_i$ of the difference settlement $s_i$ along the longitudinal unit length are represented as follows:

$$\max\Delta_i = \max\frac{|s_i - s_j|}{L_{ij}} = 0.68 \text{ mm/m}$$

Satisfy max$\Delta_i \leq [\Delta]$.

(6) The uniform load on the longitudinal beam is 151.475 kPa, and the left and right foundation stiffnesses are:

$$k_1 = \frac{1}{m}\sum_{i=1}^{m} k_i = 7.79475 \times 10^4 \text{ kN/m}^2, \; k_2 = \frac{1}{n}\sum_{i=1}^{n} k_i = 9.8215 \times 10^4 \text{ kN/m}^2$$

Thus: $\lambda_1 = \sqrt[4]{\frac{k_1 b}{4EI}} = 0.052547$; $\lambda_2 = \sqrt[4]{\frac{k_2 b}{4EI}} = 0.055672$

It can be obtained by introducing Equation (21):

$$A = q\lambda_1{}^4(\lambda_1 + \lambda_2)\left[e^{-\lambda_2 L}(\cos\lambda_2 L - \sin\lambda_2 L) - 1\right] - \frac{1}{2}q\lambda_2{}^3(\lambda_1{}^2 - \lambda_2{}^2)\left[1 - e^{-\lambda_1 L}(\cos\lambda_1 L + \sin\lambda_1 L)\right]$$
$$+ q\lambda_1{}^3 e^{-\lambda_2 L}(\lambda_1{}^2 - \lambda_2{}^2)\sin\lambda_2 L = 0.00012$$

$Q_0 =$ 5976.548 kN, $Q \leq [Q]$ meets the requirements.

## 5. Analysis of Differential Settlement Factors

It can be seen from Figure 7 that the foundation stiffness of the natural foundation is mostly of the mutant type, which better reflects the change in natural strata. Therefore, this model is selected as the basis for the calculation of settlement control standards. The average load at the bottom of the tunnel pipe is calculated via Equation (30):

$$\bar{k} \cdot s = q \tag{34}$$

Consequently:

$$L = \frac{\pi}{2} \sqrt[4]{\frac{4EI}{\bar{k}}} = \frac{\pi}{2} \sqrt[4]{\frac{4EIs}{q}} \tag{35}$$

By bringing Equations (34) and (35) and $\kappa = 0.77$ into Equation (31), the allowable average differential settlement of the longitudinal unit length $[\Delta]$ is acquired:

$$[\Delta] = \frac{2[Q]\bar{k}s^2\kappa}{\left(\bar{k}sL\kappa\right)^2 - 4[Q]^2} = \frac{2[Q]qs\kappa}{(qL\kappa)^2 - 4[Q]^2} = \frac{1.54[Q]qs}{0.29645\pi^2 q\sqrt{EIsq} - 4[Q]^2} \tag{36}$$

The width of the immersed tube is 37.95 m, its height is 11.4 m, the $EI$ is $9.7 \times 10^{10}$ kN $\cdot$ m$^2$, and the $[Q]$ is $1.6 \times 10^4$ kN. The relationship between $[\Delta]$A and $s$ under different tube bottom loads is studied, and the calculation results are shown in Figure 11.

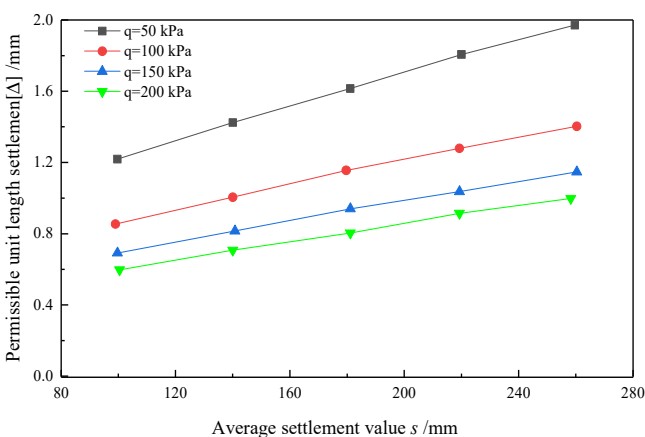

**Figure 11.** Relationship between allowable differential settlement and average settlement.

As can be seen from Figure 11, when the allowable shear stress at the joint is fixed, under the action of a specific tube bottom load, $[\Delta]$ basically increases linearly with $s$ in the range of *2L*. The larger the tube bottom $s$ is, the smaller the foundation stiffness is and the larger $[\Delta]$ is. Under different tube bottom loads, the linear relation curve between $[\Delta]$ and $s$ is a parallel line, and the correlation growth ratio is approximately stable. Therefore, the large foundation stiffness is not conducive to the control of the differential settlement. If $s$ is equal, the larger the tube bottom load is, the smaller $[\Delta]$ is, and the decreasing rate of $[\Delta]$ gradually decreases with the increase in the tube bottom load. Therefore, reducing certain additional stress has a certain control effect on the differential settlement.

The tube bottom load is selected as 150 kPa, and the relationship between $[Q]$ and $[\Delta]$ is discussed when the tube bottom $s$ is different. The calculation results are shown in Figure 12.

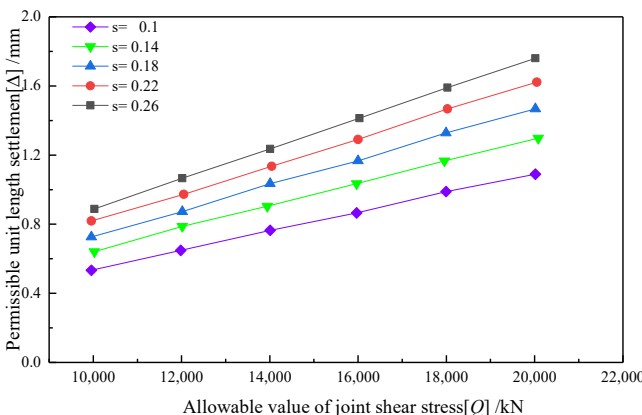

**Figure 12.** Relationship between allowable differential settlement and allowable joint shear force.

It can be seen from Figure 12 that when *s* in the study section 2*L* is a constant value, [Δ] increases with the increase in [*Q*]; that is, the larger [*Q*] is, the larger [Δ] is. When [*Q*] is a constant value, the larger *s* is, that is, the smaller the foundation length is, the larger [Δ] is, which is consistent with the conclusion in Figure 11.

## 6. Conclusions

This study conducted research on the control standards for settlement of the foundation of immersed tunnels, providing beneficial suggestions for optimizing the foundation scheme, improving construction processes, and ensuring the safety of the structural stress to the greatest extent. The main conclusions are as follows:

1. Based on Winkel's elastic foundation beam theory, a computational mechanics model of an elastic foundation beam considering the inhomogeneous soil layer was established, and the longitudinal displacement curve equation of the tunnel was obtained. Subsequently, the shear force, bending moment value, and deflection equation of the tunnel at a certain position in the inhomogeneous soil layer were calculated.

2. The infinite-length elastic foundation beam was used to simulate the immersed tube tunnel structure, and the calculation mechanics model of the elastic foundation beam considering the inhomogeneous soil layer was established. The calculation formula of the comprehensive foundation bed coefficient of layered soil foundation was derived, and the results values of different positions were determined based on the field load plate test results.

3. Combined with the inhomogeneous soil layer of computational mechanics of the elastic foundation beam model, the immersed tube longitudinal allowable differential settlement formula of unit length was deduced, which was applied to the Zhuhai immersed tube tunnel, and the relationship between the settlement control index and the related affecting factors was analyzed. Furthermore, a set of calculation methods applicable to the settlement control standard of the segmental immersed tube tunnel foundation were obtained that provide conditions for the optimization of the foundation scheme design and the improvement of construction technology.

4. The limitations of the Winkler foundation beam model include that it is only applicable to minor deformation situations and cannot provide accurate analysis results for large deformation scenarios. Moreover, the immersed tube tunnel research results presented in this study are based on the marine soils in the Hong Kong–Zhuhai–Macao region, and further verification is needed to determine whether the model is applicable to soils in other regions.

**Author Contributions:** Conceptualization, Z.L.; methodology, X.Y.; data curation, G.W.; writing—review and editing, Z.L. All authors have read and agreed to the published version of the manuscript.

**Funding:** The research presented in this paper is supported by the Fundamental Research Funds for the Central Universities (300102210115), Natural Science Basic Research Program of Shaanxi (20200-379) and Key Laboratory of Roads and Railway Engineering Safely (Shijiazhuang Tiedao University), Ministry of Education (STKF201905).

**Institutional Review Board Statement:** Not applicable.

**Informed Consent Statement:** Not applicable.

**Data Availability Statement:** Not applicable.

**Conflicts of Interest:** The authors declare no conflict of interest.

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
