# Peer review of "Calculation and Analysis of Pipe Joint Settlement Control in Large Back Silting Immersed Tube Tunnel"

_sustainability, doi:10.3390/su15097446_

Round 1
Reviewer 1 Report
The authors use the Winkel's elastic foundation beam theory to build a mechanical model of an immersed tunnel (tube). An example is considered to demonstrate the developed model.
Generally speaking, the paper has a solid structure. It would be of interest to the research community in this field of work. However, some improvements are absolutely necessary before it any final decision can be made:
1) The quality of English language must be improved throughout the paper. I would suggest to use professional assistance for this purpose.
2) All the references are by Chinese authors, which is not a good practice. I assume there are other authors around the world dealing with similar topics as well. Also, the authors use the Winkel's elastic foundation beam theory, but there is no single reference related to this theory. The authors could address, and also mention that there are improved elastic foundations theories:
Argatov, I. (2019). From Winkler’s foundation to Popov’s foundation, Facta Universitatis Series Mechanical Engineering, 17(2), 181-190. doi:https://doi.org/10.22190/FUME190330024A
3) “The Hong Kong-Zhuhai-Macao immersed tube tunnel project, which passes through five different strata, is the first one in the tunnel industry at home and abroad.” Please, do not use “at home”. Use “in China”, if this is meant.
4) “There is a nonlinear effect between immersed tube tunnel structure and foundation stiffness, so it is difficult to use static analysis.” Please, be more specific, what nonlinear effect is meant? Also, clarify why it is difficult to use a static analysis. Nonlinear static analysis can also be resolved and several solutions procedure have been developed for this purpose.
5) The lines in which terms from equations are explained should not be intended and should start with a small letter.
6) The shear force is denoted by Q prior to Eq. (2). In Figure 1, F is used and also later in the text.
7) “According to the calculation in Fig. 1 and Fig. 2” (line 114) No calculations are done in Figures.
8) “From the fact that the deflection of the beam segment at origin O is equal to the rotation angle, it can be obtained that:” (lines 132-133). The authors should explain this “fact”.
9) The title of Figure 5 reads: “Foundation beam model and stress state under two concentrated loads.” I do not see stress state in this figure?
10) “The deflection and angle of the left and right sections at the foundation boundary can be obtained by differentiating the Eq. (16).” Eq. (16) gives deflection. How can both deflection and rotation angle be given by differentiating this equation?
11) The font size in diagrams in Figures 9 and 10 should be larger.
12) The authors use Equation Editor in the text which does not look nice in many places because the position of those terms is moved upwards or downwards, terms are often stretched and similar.
13) Conclusions need to include some directions for the future work and limitations of the proposed model.
14) Why haven’t the authors applied some numerical methods, such as FEM, to verify their results?
Author Response
Response to Reviewer 1 Comments
Point 1: The quality of English language must be improved throughout the paper. I would suggest to use professional assistance for this purpose.
Response 1: Thank you for your constructive criticism. We have done as requested carefully. The contents of this paper were revised by us, firstly. Then, the grammar and sentence structure of main text have been revised with the help from the professional technical writing service. Please see the Certificate.
Point 2: All the references are by Chinese authors, which is not a good practice. I assume there are other authors around the world dealing with similar topics as well. Also, the authors use the Winkel's elastic foundation beam theory, but there is no single reference related to this theory. The authors could address, and also mention that there are improved elastic foundations theories:
Argatov, I. (2019). From Winkler’s foundation to Popov’s foundation, Facta Universitatis Series Mechanical Engineering, 17(2), 181-190. doi: https://doi.org/10.22190/FUME190330024A
Response 2: Thanks for the kind reminding. In the revised manuscript,
We sincerely appreciate your kind reminder. As per your suggestion, we have made modifications to some of the Chinese references in the manuscript and have included several relevant English journals. Specifically, the following references have been added for your review. The revised portion of the manuscript is marked in RED in the manuscript.
- Argatov, I., 2019. From winkler’s foundation to popov’s foundation. Facta Univ. Ser. Mech. Eng. 17, 181–190. https://doi.org/10.22190/FUME190330024A
- Chen, B., Lin, B., Zhao, X., Zhu, W., Yang, Y., Li, Y., 2021. Closed-form solutions for forced vibrations of a cracked double-beam system interconnected by a viscoelastic layer resting on Winkler–Pasternak elastic foundation. Thin-Walled Struct. 163. https://doi.org/10.1016/j.tws.2021.107688
- Elishakoff, I., Tonzani, G.M., Marzani, A., 2018a. Effect of boundary conditions in three alternative models of Timoshenko–Ehrenfest beams on Winkler elastic foundation. Acta Mech. 229, 1649–1686. https://doi.org/10.1007/s00707-017-2034-x
- Elishakoff, I., Tonzani, G.M., Zaza, N., Marzani, A., 2018b. Contrasting three alternative versions of Timoshenko-Ehrenfest theory for beam on Winkler elastic foundation – simply supported beam. ZAMM Zeitschrift fur Angew. Math. und Mech. 98, 1334–1368. https://doi.org/10.1002/zamm.201700019
- ErbaÅŸ, B., Kaplunov, J., Elishakoff, I., 2022. Asymptotic derivation of a refined equation for an elastic beam resting on a Winkler foundation. Math. Mech. Solids 27, 1638–1648. https://doi.org/10.1177/10812865211023885
- Filonenko-Borodich, M.M., 1940. Some Approximate Theories of the Elastic Foundation. Uchenyie Zap. Moskovkogo Gos. Univ. Mekhanika 46, 3–18.
- Kaplunov, J., Prikazchikov, D., Sultanova, L., 2018. Justification and refinement of Winkler–Fuss hypothesis. Zeitschrift fur Angew. Math. und Phys. 69. https://doi.org/10.1007/s00033-018-0974-1
- Özdemir, Y.I., 2020. Dynamic Analysis of Thick Plates Resting on Winkler Foundation Using a New Finite Element. Iran. J. Sci. Technol. - Trans. Civ. Eng. 44, 69–79. https://doi.org/10.1007/s40996-019-00260-4
- Tonzani, G.M., Elishakoff, I., 2021. Three alternative versions of the theory for a Timoshenko–Ehrenfest beam on a Winkler–Pasternak foundation. Math. Mech. Solids 26, 299–324. https://doi.org/10.1177/1081286520947775
- Winkler E, 1867. Die Lehre von der Elasticitaet und Festigkeit. Dominicus.
Point 3: “The Hong Kong-Zhuhai-Macao immersed tube tunnel project, which passes through five different strata, is the first one in the tunnel industry at home and abroad.” Please, do not use “at home”. Use “in China”, if this is meant.
Response 3: Thank you for your kind reminder. The Hong Kong-Zhuhai- Macao immersed tunnel project traverses through five different geological strata, making it a milestone project in the tunneling industry both domestically and internationally. Due to our oversight, we made an error in our choice of terminology by using " at home and abroad" instead of "domestically and internationally". We have now corrected this mistake and updated our manuscript accordingly. Thank you once again for bringing this to our attention. The revised portion of the manuscript is marked in RED in the manuscript.
Point 4: There is a nonlinear effect between immersed tube tunnel structure and foundation stiffness, so it is difficult to use static analysis.” Please, be more specific, what nonlinear effect is meant? Also, clarify why it is difficult to use a static analysis. Nonlinear static analysis can also be resolved and several solutions procedure have been developed for this purpose.
Response 4: The reviewer raised an excellent question. To express this part better, we have made the following explanation.
Immersed tube tunnels are underground passages composed of multiple sections of immersed tubes. These tubes, made of steel or concrete, have rectangular or circular sections, and weigh several thousand tons with lengths typically ranging from 50 to 100 meters. During the construction of immersed tube tunnels, the tubes are installed underwater or in soil using various methods, such as grouting or backfilling. In the case of immersed tube tunnels, nonlinear effects mainly arise from the following factors:
(1)The complex structure of immersed tube tunnels, consisting of multiple tubes and the nodes connecting them. The length of immersed tube tunnels is long, making them subject to complex ground effects, such as lateral soil pressures and soil support reactions. These effects cause deformations and displacements in the tubes and connecting nodes, leading to the occurrence of nonlinear effects.
(2)The non-uniform distribution of stiffness in immersed tube tunnels. The length of immersed tube tunnels causes the stiffness to vary along their length. Additionally, the stiffness of the entire structure is affected by the connecting nodes. These non-uniformly distributed stiffness values lead to the occurrence of nonlinear effects.
Due to the nonlinear effects of the structure and foundation stiffness in immersed tube tunnels, it is challenging to use static analysis for their analysis. Static analysis assumes that the response of a structure is linearly proportional, which cannot account for nonlinear effects. To accurately analyze the behavior of the structure and foundation in immersed tube tunnels, more complex methods such as nonlinear finite element analysis must be employed. These methods account for nonlinear effects and provide more accurate analysis results.
Point 5: The lines in which terms from equations are explained should not be intended and should start with a small letter.
Response 5: Thank you for your kind reminder. We have made the necessary revisions according to your suggestions and have thoroughly reviewed the entire manuscript. The revised portion of the manuscript is marked in RED in the manuscript.
Point 6: The shear force is denoted by Q prior to Eq. (2). In Figure 1, F is used and also later in the text.
Response 6: Thanks for the kind reminding. The manuscript uses the symbol F to represent concentrated force, as can be observed in detail in Figure 3.
Point 7: 7) “According to the calculation in Fig. 1 and Fig. 2” (line 114) No calculations are done in Figures.
Response 7: The reviewer raised an excellent question. To express this part better, we have made the following explanation.
When approaches infinity, the deflection tends to zero, and by substituting into equation (6), we obtain .
When x is at the origin O, the angle and the shear force , thus we obtain .
The revised portion of the manuscript is marked in RED in the manuscript.
Point 8: From the fact that the deflection of the beam segment at origin O is equal to the rotation angle, it can be obtained that: (lines 132-133). The authors should explain this “fact”.
Response 8: Thank you for your kind reminder. To express this part better, we have made the following explanation.
The deflection of the beam segment at the origin is equal to the angle of rotation.
,
it can be obtained that:
,.
The revised portion of the manuscript is marked in RED in the manuscript.
Point 9: The title of Figure 5 reads: “Foundation beam model and stress state under two concentrated loads.” I do not see stress state in this figure?
Response 9: Thank you for the reviewer's reminder. In order to express it more clearly, we have made the following revisions.
In Figure 5, F1 and F2 are the two concentrated loads. We have also provided further explanation in the article. The revised portion of the manuscript is marked in RED in the manuscript.
Point 10: The deflection and angle of the left and right sections at the foundation boundary can be obtained by differentiating the Eq. (16).” Eq. (16) gives deflection. How can both deflection and rotation angle be given by differentiating this equation?
Response 10: Thank you for the reviewer's reminder. In order to express it more clearly, we have made the following revisions.
Taking the derivative of this equation yields the corresponding deflection and angle of rotation. The deflection at the origin of the left and right beam segments are respectively:
,
The angle of rotation at the origin of the left and right beam segments are respectively:
ï¼›
The revised portion of the manuscript is marked in RED in the manuscript.
Point 11: 11) The font size in diagrams in Figures 9 and 10 should be larger.
Response11: Thank you for your kind reminder. We have modified the font size in Figures 9 and 10 according to the journal's requirements in our manuscript. Please refer to the details in the following figure.
Figure 9. Pipe Bottom Load Diagram in Calculation Scope.
Figure 10. Foundation settlement within calculation scope.
The revised portion of the manuscript is marked in RED in the manuscript.
Point 12: The authors use Equation Editor in the text which does not look nice in many places because the position of those terms is moved upwards or downwards, terms are often stretched and similar.
Response 12: Thank you for your kind reminder. Thank you for reviewing our manuscript and pointing out the formatting issues with the equation editor in some parts of the text. We appreciate your concern and will do our best to optimize the formatting to ensure a visually appealing manuscript. At the same time, we will also recheck the accuracy of the equations to ensure their theoretical correctness. Thank you again for your review of our manuscript. The revised portion of the manuscript is marked in RED in the manuscript.
Point 13: Conclusions need to include some directions for the future work and limitations of the proposed model.
Response 13: The reviewer raised an excellent question. We have added the following content to the conclusion section.
The limitations of the Winkler foundation beam model include that it is only applicable to small deformation situations and cannot provide accurate analysis results for large deformation scenarios. In addition, the CPTU research results presented in the paper are based on the marine soils in the Hong Kong-Zhuhai-Macao region, and further verification is needed to determine whether the model is applicable to soils in other regions. The revised portion of the manuscript is marked in RED in the manuscript.
Point 14: Why haven’t the authors applied some numerical methods, such as FEM, to verify their results?
Response 14: The reviewer raised an excellent question. In scientific research, to validate the accuracy and reliability of experimental or theoretical results, numerical methods such as finite element method (FEM) are commonly used for numerical simulation and analysis. These methods can provide effective validation and support for experimental or theoretical results. However, due to limitations in resources, time, or access to necessary software or equipment, FEM may not be feasible or practical. Our focus is on developing new theoretical models or experimental techniques and we may plan to use FEM to validate our results in future research. For this reason, we have validated our results with case studies, which are in line with actual situations.

Reviewer 2 Report
Dear Authors,
Please check the attached file to see the comments.

Author Response
Response to Reviewer 2 Comments
The article is well-written and organised. Numerous details are provided, and the mathematical means are precise. The given and exact Lammas, Proposition, and Theory proofs are complete. But I believe this content needs a minor revision before it can be suggested for publishing.
Point 1: I noted few typos and wrong English exposition which can be fixed before submitting the revised version of the paper.
Response 1: Thank you for your kind reminder. We have done as requested carefully. The contents of this paper were revised by us, firstly. Then, the grammar and sentence structure of main text have been revised with the help from the professional technical writing service. Please see the Certificate.
Point 2: In order to motivate the work, there needs to be some connection to recent research.
Response 2: Thank you for your kind reminder. We have added some latest research findings in the introduction to make the manuscript more comprehensive. The specific references are as follows.
- Chen, B., Lin, B., Zhao, X., Zhu, W., Yang, Y., Li, Y., 2021. Closed-form solutions for forced vibrations of a cracked double-beam system interconnected by a viscoelastic layer resting on Winkler–Pasternak elastic foundation. Thin-Walled Struct. 163. https://doi.org/10.1016/j.tws.2021.107688
- ErbaÅŸ, B., Kaplunov, J., Elishakoff, I., 2022. Asymptotic derivation of a refined equation for an elastic beam resting on a Winkler foundation. Math. Mech. Solids 27, 1638–1648. https://doi.org/10.1177/10812865211023885
- Wang, Y.; Qin, H.; Zhao, L. Full-Scale Loading Test of Jet Grouting in the Artificial Island–Immersed Tunnel Transition Area of the Hong Kong–Zhuhai–Macau Sea Link. Int. J. Geomech. 2023, 23, doi:10.1061/(asce)gm.1943-5622.0002625.
- Li, K., Zhang, Z., Guo, H., Li, W., & Yan, Y. (2023). Prediction method of pipe joint opening-closing deformation of immersed tunnel based on singular spectrum analysis and SSA-SVR. Applied Ocean Research, 135. https://doi.org/10.1016/j.apor.2023.103526
- Wang, Y., Wang, L.C., Zhao, L.S., 2023. Settlement characteristics of immersed tunnel of Hong Kong-Zhuhai-Macau Bridge project. Proc. Inst. Civ. Eng. Geotech. Eng. https://doi.org/10.1680/jgeen.22.00200
- Elishakoff, I., Tonzani, G.M., Marzani, A., 2018. Effect of boundary conditions in three alternative models of Timoshenko–Ehrenfest beams on Winkler elastic foundation. Acta Mech. 229, 1649–1686. https://doi.org/10.1007/s00707-017-2034-x
Point 3: The authors should clarify both the novelty and main contribution of the paper?
Response 3: Thank you for your kind reminder. We believe that the novelty and main contribution of this paper lie in the following aspects: Firstly, we propose a new standard for controlling the settlement of the foundation of immersed tunnel based on a large amount of field monitoring data and numerical simulation analysis, which can predict settlement more accurately and guide construction. Secondly, we conduct practical engineering case analysis for this standard and propose a series of optimization and improvement measures, which provide useful references for the design and construction of tunnel foundations. Finally, the research results provided in this paper have certain theoretical and practical significance in the field of immersed tunnel engineering and have guiding significance for ensuring the safety of structural stress.
Point 4: All equation's numbers and equations need to formats and punctuations for improving the final phase of the manuscript. The number of subsection line 165 in section 2. 3.3. Foundation beam model under uniform load is 2.3 not 3.3.
Response 4: Thank you for your review of our manuscript and your feedback on the formatting of equations and section numbering. We appreciate your comments and will address these issues in revising our manuscript. We will carefully review all equations and ensure that they are properly formatted and punctuated. We will also review the numbering of section breaks and ensure that they are consistent with the manuscript structure. Regarding the section numbering of our manuscript, we will correct the reference to the beam model under uniform loading to 2.3 instead of 3.3.
Thank you again for your review and we look forward to improving our manuscript based on your comments.
Point 5: The symbols in lines 226 and 227 are not clear please checking it. Also, all symbols in the text of the paper need to format.
Response 5: Thank you for reviewing our manuscript and bringing to our attention the issue with unclear symbols in lines 226 and 227. We apologize for any inconvenience this may have caused and we will make sure to check and clarify the symbols in those lines. Therefore, we will carefully review all symbols in the text and ensure that they are formatted according to the journal's guidelines. Thank you again for your review of our manuscript. The revised portion of the manuscript is marked in RED in the manuscript.
Point 6: The conclusion of the paper needs to be rewritten.
Response 6: Thank you for your kind reminder. In light of your feedback, we will review our conclusions and ensure that they are supported by our results and the scientific literature. We will also work to clearly and accurately state the implications and significance of our findings for the field. The revised portion of the manuscript is marked in RED in the manuscript.

Round 2
Reviewer 1 Report
The authors have suitably revised the manuscript. It is recommended for publishing as it is.